# Effect of daily physical activity on ambulatory blood pressure in pregnant women with chronic hypertension: A prospective cohort study protocol

**Yanxiang Lv**[◉], **Rui Hu**[◉], **Yan Liang**[◉], **Ying Zhou**[‡], **Yanan Lian**[‡], **Tongqiang He**[◉]*

Department of Obstetrics and Gynecology Intensive Care Unit, Northwest Women's and Children's Hospital, Xi'an City, Shaanxi Prov, China

◉ These authors contributed equally to this work.
‡ YZ and YL also contributed equally to this work.
* xbfymicu@163.com

**Data Availability Statement:** No datasets were generated or analysed during the current study. All relevant data from this study will be made available upon study completion.

## Abstract

### Background

Physical activity, a first-line approach for the treatment of non-gestational hypertension globally, has been shown to benefit most pregnant women in many respects. The benefits and risks of prenatal physical activity in complicated pregnancies, such as preeclampsia and chronic hypertension, require further investigation. It is worth conducting studies to address questions about physical activity during pregnancy in women with chronic hypertension, such as the benefits and risks, frequency, duration, and intensity. This prospective cohort study aims to investigate whether moderate-intensity daily physical activity reduces ambulatory blood pressure in pregnant women with chronic hypertension.

### Methods

Pregnant women with chronic hypertension at $11^{+0}$ to $13^{+6}$ gestational weeks will be recruited from the outpatient clinic and divided into moderate- and light-intensity physical activity groups according to the intensity of the 7-day physical activity monitored using the model wGT3X-BT accelerometer. 24-h ambulatory blood pressure monitoring will be performed at enrollment as a baseline and will be repeated in the second and third trimesters. The primary outcome is the difference in the change in 24-h ambulatory systolic blood pressure from the first to the third trimester between the groups. Secondary outcomes include the difference of change in other ambulatory (24-h diastolic, daytime, and nighttime) and office blood pressure variables from the first to the second and third trimesters, the incidence of severe hypertension ($\geq$160/110 mmHg), and changes in the type and dosage of antihypertensive medication. The primary and secondary outcomes related to changes in blood pressure from baseline to the second and third trimesters between the groups will be analyzed using Student's independent t-test or the Mann–Whitney U test.

**Funding:** This project is supported by the Key Research and Development Program of Shaanxi China (Program No. 2022SF-277) and the Scientific Research Project of Northwest Women's and Children's Hospital (Program No. 2018LQ03, 2022-YN10). The funders have no role in study design, data collection and analysis, the decision to publish, or the preparation of the manuscript. No author receives a salary from any of the funders.

**Competing interests:** The authors have declared that no competing interests exist.

**Abbreviations:** ABPM, Ambulatory Blood Pressure Monitoring; CPM, counts per minute; DBP, diastolic blood pressure; HDP, hypertensive disorders of pregnancy; LPA, light-intensity physical activity; MPA, moderate-intensity physical activity; SBP, systolic blood pressure; SGA, small for gestational age.

## Discussion

This cohort study will provide a basis for randomized controlled trials and verify an easily achieved, economical, and non-fetotoxic approach for adjuvant blood pressure management in pregnant women with chronic hypertension.

## Registry

This study is registered with the Chinese Clinical Trials Registry (NO. ChiCTR2200062094). Date Registered: 21/07/2022.

## 1. Introduction

Chronic hypertension affects 0.3%–4.3% of pregnant women, and the risk of preeclampsia is 5.43 times greater in women with chronic hypertension than in those with normotension [1] and may evolve into eclampsia, which is serious and challenging to treat [2]. Furthermore, chronic hypertension is associated with a higher risk of cesarean section, maternal mortality, preterm birth, stillbirth, small for gestational age (SGA), low birth weight, and neonatal intensive unit admission [1, 3]. The above-mentioned adverse maternal and perinatal outcomes are more likely to occur in chronic hypertension-superimposed preeclampsia than in preeclampsia alone [4, 5]. Extensive research has been conducted to identify risk factors and biomarkers of preeclampsia and the corresponding adverse outcomes for prediction and prevention [6–9]. Blood pressure management is crucial throughout the perinatal period for pregnant women with chronic hypertension, whether or not the condition progresses into preeclampsia. Two large randomized trials [10, 11] found that antihypertensive treatment targeting a blood pressure of < 140/90 mmHg in chronic hypertension was safe and beneficial for the pregnant women and infant. However, some medications should be avoided during pregnancy due to possible or proven concerns regarding fetotoxicity [12–14]. Pregnant women have limited options for antihypertensive medications compared with those of non-pregnant women. Pregnancy termination should be considered to avoid maternal and fetal complications such as placental abruption and cardiovascular and cerebrovascular adverse events in women with uncontrolled hypertension after active intervention. Lifestyle modification can enhance the efficacy of antihypertensive therapies. Guidelines for gestational hypertension in pregnant and non-pregnant women [15–17] recommend lifestyle modification, including salt reduction, smoking and alcohol cessation, healthy diet, weight reduction, and regular physical activity.

Exercise is recommended as a first-line adjunct to antihypertensive treatment during the non-gestational period. A systematic review [18] showed that physical activity can reduce blood pressure in adults with normal blood pressure, prehypertension, and hypertension. A two-center, single-blinded randomized clinical trial [19] showed that moderate-intensity aerobic exercise lasting 12 weeks reduced 24-h and daytime ambulatory blood pressure and office systolic blood pressure (SBP) in patients with resistant hypertension. Furthermore, the reduction of SBP is correlated with a lower risk of cardiovascular morbidity and all-cause mortality [20]. Physical activity during pregnancy benefits most women in many aspects. A secondary analysis of a prospective cohort study [21] revealed that patients who were more physically active during pregnancy had a shorter duration of active labor. A series of studies [22–25] showed that physical activity reduced the risk of adverse maternal and neonatal outcomes, as well as the risk of excessive gestational weight gain, gestational diabetes, preeclampsia, and preterm birth. Psychological benefits [26] of physical activity include reduced stress, anxiety,

fatigue and depression, and improved well-being. Pregnant women without contraindications are advised to maintain moderate-intensity physical activity for accumulating at least 150 min for ≥3 days per week. Although uncontrolled hypertension and gestational hypertension are considered absolute and relative contraindications to physical activity, guidelines [27–29] point out that the benefits and risks of prenatal physical activity in complicated pregnancies, such as preeclampsia, gestational hypertension, and chronic hypertension, are also needed for further identification. An observational cohort study suggested that the risk of chronic hypertension after hypertensive disorders of pregnancy might be markedly reduced by adherence to a beneficial lifestyle; however, there was no clear evidence of effect modification by physical activity on the association between hypertensive disorders of pregnancy and chronic hypertension [30]. Two studies [31, 32] conducted in 116 pregnant women with chronic hypertension and/or previous preeclampsia showed that physical exercise using a stationary bicycle once a week did not interfere with the mode of delivery, maternal or neonatal morbidity, changes in blood pressure or heart rate. Existing studies are limited and insufficient to guide pregnancy care for women with chronic hypertension. It is worth conducting studies to explore and address questions about physical activity during pregnancy in women with chronic hypertension, such as the benefits and risks, frequency, duration, and intensity.

We hypothesize that moderate-intensity physical activity is a low-cost, safe, and effective adjunct to blood pressure management in pregnant women with chronic hypertension. This prospective cohort study aims to investigate whether moderate-intensity daily physical activity reduces ambulatory blood pressure in pregnant women with chronic hypertension. This study provides a basis for randomized controlled trials and an easily achieved, economical, safe, and non-fetotoxic approach to adjuvant blood pressure management in pregnant women with chronic hypertension.

## 2. Objectives

### Primary objective

To investigate whether moderate-intensity daily physical activity reduces ambulatory blood pressure in pregnant women with chronic hypertension.

### Secondary objectives

To evaluate the effects of daily physical activity on the incidence of preeclampsia and adverse perinatal outcomes in pregnant women with chronic hypertension.

To assess the effect of physical activity on blood pressure in pregnant women with preeclampsia superimposed upon chronic hypertension.

## 3. Materials and methods

The current study is a part of the project titled "Effect of Physical Activity in Leisure Time on Ambulatory Blood Pressure Among Pregnant Women at High Risk for Preeclampsia", registered on the Chinese Clinical Trials Registry (No. ChiCTR2200062094). Date registered: 07/21/2022. This study adheres to the guidelines of the Standard Protocol Items: Recommendations for Interventional Trials (SPIRIT) provided by the Enhancing the Quality and Transparency of Health Research (EQUATOR) Network.

### 3.1 Study design

This prospective cohort study with a parallel two-arm group will be performed at the outpatient clinic of the Northwest Women's and Children's Hospital (NWCH), a specialized tertiary

hospital for women and children in Shaanxi Province, China. Patient recruitment will be conducted from August 2022 to December 2023, and each patient will be followed up until pregnancy termination. The complete date range for participant recruitment, follow-up, data collection is scheduled between August 2022 and June 2024. The recruited pregnant women with chronic hypertension will be divided into moderate-intensity physical activity (MPA) and light-intensity physical activity (LPA) groups according to the intensity of daily physical activity in the first trimester. Ambulatory blood pressure and daily physical activity will be monitored during the first, second, and third trimesters.

### 3.2 Inclusion criteria

The inclusion criteria are as follows: maternal age $\geq$ 18 years old, singleton pregnancy and fetal survival at $11^{+0}$ to $13^{+6}$ weeks of gestational age, history of chronic hypertension defined as hypertension diagnosed before pregnancy or at the first prenatal visit.

### 3.3 Exclusion criteria

Pregnant women with one of the following conditions will be excluded: severe cardiovascular or respiratory disease, hyperthyroidism, pregestational diabetes mellitus or gestational diabetes, incompetent cervix, recurrent spontaneous abortion, a history of spontaneous preterm birth, threatened abortion or inevitable abortion, placenta previa, severe anemia, malnutrition, or very low body weight (body mass index $< 12 \text{kg/m}^2$), severe mental disorders, inability to express their will, other obvious abnormal signs, laboratory examination, and other clinical diseases.

### 3.4 Procedures

Pregnant women with chronic hypertension will be invited to participate at $11^{+0}$ to $13^{+6}$ weeks of gestational age. A 7-day maternal physical activity evaluation will be performed for grouping once they have consented to participate in the study. Simultaneously, 24-h ambulatory blood pressure monitoring will be conducted as a baseline. Participants will be scheduled for a 7-day physical activity evaluation and ambulatory blood pressure monitoring in the second ($20^{+0}$ to $21^{+6}$ gestational weeks) and third trimesters ($30^{+0}$ to $31^{+6}$ gestational weeks). Fig 1 shows a timeline of the procedures and evaluations.

**Demographics, medical history, and medication.** Demographic information, participation in pre-pregnancy physical activity, and medical history will be recorded. The obstetrician will prescribe examination items corresponding to the gestational week. A physical examination will be performed at every visit to exclude any limitations in physical activity or conditions that may require participant exclusion from the study. All medical data and therapies will be documented. Participants will be instructed about home blood pressure monitoring and recording. Oral antihypertensive drugs will be administered according to the blood pressure at the time of enrollment and the dose of existing medication. A blood pressure threshold of 140/90 mmHg will be used to initiate and titrate antihypertensive therapy, and the target ranges should be 120–139 mmHg for SBP and 80–89 mmHg for diastolic blood pressure (DBP). The initial antihypertensive management should be monotherapy, with labetalol as the first choice. Changes in the dosage and type of antihypertensive medications should be recorded. All participants will be instructed on salt-restricted diet, sleep schedule, oral aspirin of 100 mg/day from enrollment until 36 gestational weeks, and calcium supplements $> 1$ g/day.

**Ambulatory blood pressure monitoring.** During each assessment period (first, second, and third trimesters), 24-h ambulatory blood pressure measurements will be monitored using the Ri-cardio (Rudolf Riester GmbH, Jungingen, Germany). Measurements will be taken while

| Procedures and Evaluations | Baseline | Monitoring | | Closeout |
|---|---|---|---|---|
| | $t_0$ (11$^{+0}$ to 13$^{+6}$ gestational weeks) | $t_1$ (20$^{+0}$ to 21$^{+6}$ gestational weeks) | $t_2$ (30$^{+0}$ to 31$^{+6}$ gestational weeks) | $t_3$ (Delivery) |
| *Eligibility screen* | X | | | |
| *Informed consent* | X | | | |
| *Demographic information* | X | | | |
| *Medical history* | X | | | |
| *Physical examination* | X | X | X | X |
| *Obstetrical examination items* | X | X | X | X |
| *Counseling of lifestyle and medication* | X | X | X | X |
| *Daily physical activity monitoring* | X | X | X | |
| *Ambulatory blood pressure monitoring* | X | X | X | |
| *Compliance with medications* | | X | X | |
| *Borg Perceptual Motor Intensity Scale* | X | X | X | X |
| *Adverse events* | X | X | X | X |
| *Maternal and Fetal outcomes* | | X | X | X |

**Fig 1. Scheduling of study procedures and evaluations.**

monitoring the non-dominant arm every 30 min during the day and every 60 min at night. The participants will be instructed to proceed with their normal daily activities during the measurements. All participants will keep a record of their wakeup, bedtime, medications intake, and any other important events. The ambulatory blood pressure monitoring reports will be considered successful if 80% of all readings are available. Resting office blood pressure will be measured by trained nurses using validated devices following the recommendations of the European Society of Cardiology/European Society of Hypertension, with appropriate cuffs selected according to individual-sized arms.

**Daily physical activity monitoring.** Daily physical activity will be monitored using a wGT3X-BT accelerometer (ActiGraph, Florida, USA). Participants will be instructed to wear the accelerometer on the nondominant wrist for 7 days during each assessment period (first, second, and third trimesters). The recordings will be considered valid if worn for more than 10 h daily and includes data for at least two working days (Monday to Friday) and one rest day (Saturday to Sunday). Data from the accelerometer will be downloaded, and raw data .csv file extraction will be performed using Actilife 6.1 Software. The analyzed data will provide the average daily acceleration and express physical activity as the total movement volume. The Freedson equation [33] was used to derive the activity intensity, duration, and frequency from the measured activity in counts per minute (CPM) per day as follows: very vigorous (>9498 CPM), vigorous (5725–9498 CPM), moderate (1952–5724 CPM), light (100–1952 CPM), and

sedentary (<100 CPM). Therefore, in this study, LPA and MPA are defined as physical activities ranging from 100 to 1952 CPM and ≥1952 CPM, respectively. The Borg Perceptual Motor Intensity Scale records ratings of perceived exertion.

**Endpoint events.** The endpoint events include delivery, vaginal bleeding, regular contractions, premature rupture of membranes, headache, dizziness, dyspnea, chest pain, muscle weakness affecting balance, and other severe clinical diseases.

## 3.5 Outcomes

**Primary outcome.** The primary outcome will be the difference in the change in 24-h ambulatory SBP from the first to the third trimester between the groups.

**Secondary outcomes.**

- Difference in the change in 24-h ambulatory SBP from the first to the second trimester between the groups.

- Differences in the changes in other ambulatory blood pressure parameters (24-h diastolic, daytime, and nighttime) from the first to the second and third trimesters between the groups.

- Differences in the change in office blood pressure from the first trimester to the second and third trimesters between the groups.

- The incidence of severe hypertension (≥160/110 mmHg).

- The change in the type and dosage of antihypertensive medication.

**Exploratory outcomes.** The incidence of preeclampsia and the effect of physical activity on blood pressure in pregnant women with preeclampsia superimposed upon chronic hypertension are considered exploratory outcomes. The incidence of adverse maternal and fetal outcomes, including eclampsia, miscarriage, preterm birth, stillbirth, SGA, haemolysis, elevated liver enzymes, and low platelet syndrome, will be determined. Adverse events in the groups, including activity tolerance decline, chest tightness, shortness of breath, and the incidence of heart failure, will be recorded.

## 3.6 Blinding of study

This study will be conducted in a double-blinded manner. Until the completion of the study, neither the investigators nor the participants will know the results of their physical activity.

## 3.7 Study status

The protocol version number is V1.0, dated April 2022. Patient recruitment will be conducted between August 2022 and December 2023, and each patient will be followed up until pregnancy termination.

## 3.8 Ethics and consent to participate

This study has been approved by the Ethics Committee of NWCH (approval number:22–028). Written informed consent will be obtained from all participants before enrollment. None of the authors will have access to information that could identify the individual participants during or after data collection.

### 3.9 Sample size and statistical analysis

The study is powered by the primary outcome measure of the change in 24-h ambulatory SBP from the first to the third trimester. The sample size calculation is based on the results of Lopes S et al. [19], who showed a decrease of -7.3 ± 12.7 mmHg in 24-h ambulatory SBP in the exercise group compared with 1.1 ± 8.2 mmHg in the control group. A sample of 72 pregnant women (36 per group), with a two-sided significance level of 0.05, provide 90% statistical power to demonstrate this difference in 24-h ambulatory SBP. To accommodate a 20% attrition rate, 90 pregnant women will be recruited (PASS11, independent *t*-tests; allocation ratio = 1).

Continuous variables will be presented as means ± standard deviation, or median (interquartile range) according to a normal distribution. Between-group differences at baseline will be tested using Student's independent t-test or the Mann–Whitney U test. Categorical variables will be expressed as counts and percentages. Categorical variables will be tested between groups at baseline using the chi-square test or Fisher's exact test, if appropriate. The assessment of compliance with daily physical activity and safety will be summarized for the two groups. A cutoff of 80% baseline physical activity during the second and third trimesters will be used to determine whether compliance well. Adverse events, if any, will be reported. The primary and secondary outcomes associated with changes in blood pressure from the first to the second and third trimesters between groups will be analyzed using the Student's independent t-test or the Mann–Whitney U test. The analyses will be conducted based on the intention-to-treat principle. The intention-to-treat population will comprise all patients, except for those who do not complete ambulatory blood pressure measurement and physical activity monitoring at baseline or in the third trimester for any reason, such as miscarriage or preterm birth. The per-protocol set will be used for sensitivity analysis. The per-protocol set will comprise patients who complete ambulatory blood pressure measurements and physical activity monitoring at three time points with good compliance. Relative risks and 95% confidence intervals will also be provided. Statistical significance will be set at $p < 0.05$. The data will be analyzed using SPSS 25 statistical software.

## 4. Discussion

This prospective cohort study is innovative in exploring lifestyle interventions for adjuvant blood pressure management in pregnant women with chronic hypertension. Lifestyle modification may enhance the effects of antihypertensive treatment, reduce the dosage and type of blood pressure medication, and avoid the incidence of severe hypertension ($\geq$160/110 mmHg). Lifestyle modification, most of which is easily achieved but slowly efficacious, is an economical, safe, and non-fetotoxic approach for adjuvant blood pressure management. However, research on the effects of exercise interventions on pregnant women with chronic hypertension is limited. Increasing evidence suggests that exercise may act as a potential epigenetic trigger for cardiovascular diseases [34]. Studies have shown that epigenetic changes such as microRNA expression, histone posttranslational modifications, and DNA methylation may play key roles in preeclampsia [35–37]. These findings provide the theoretical foundation for our research.

Physiological responses to physical activity are greater during pregnancy than that during non-gestational and become more pronounced as gestational age increases, mainly including changes in cardiac output, heart rate, and ventilatory function. Hemodynamic changes during pregnancy in women with hypertension are substantially different from those in normotensive women, particularly in terms of cardiac output and vascular resistance [38]. Therefore, the intensity and duration of physical activity interventions should be more cautious, and more

attention should be paid to cardiorespiratory fitness in pregnant women with chronic hypertension. This cohort study aims to explore the safe and effective intensity of physical activity and provide a basis for randomized controlled trials in pregnant women with chronic hypertension. No intervention will be applied to the participants; instead, pregnant women will be grouped into LPA and MPA groups according to the intensity of physical activity monitored by the wGT3X-BT accelerometer during the first trimester.

Physiological changes in the cardiovascular system during pregnancy benefit the uteroplacental circulation. Systemic vascular resistance begins to decline at 6- to 8-week gestational age, reaches a nadir at around 18 to 20 weeks, and rises again in the late second and early third trimesters. This decrease in systemic vascular resistance is correlated with a decrease in blood pressure. In addition, the effect of physical activity on blood pressure is slow, for instance moderate-intensity aerobic exercise lasting 12 weeks reduces 24-h and daytime ambulatory blood pressure in patients with resistant hypertension [19]. Based on the dynamic process of blood pressure and the slow efficacy of physical activity, primary and secondary outcomes in this study will be measured at $11^{+0}$ to $13^{+6}$ gestational weeks as baseline, $20^{+0}$ to $21^{+6}$ gestational weeks, and $30^{+0}$ to $31^{+6}$ gestational weeks, respectively. The Control of Hypertension in Pregnancy Study (CHIPS) and the Chronic Hypertension and Pregnancy (CHAP) trial verified that targeting a blood pressure of $< 140/90$ mmHg was associated with better maternal and neonatal outcomes, with no increased risk of SGA. Therefore, a blood pressure threshold of 140/90 mmHg should be used to initiate and titrate antihypertensive therapy, and the target ranges should be 120–139 mmHg for SBP and 80–89 mmHg for DBP. Although maintaining the same antihypertensive medications throughout the study period makes the design of the study more reasonable, it is not ethically or medically feasible. Therefore, the incidence of severe hypertension and changes in antihypertensive medication use were considered secondary outcomes in this study. The results will be presented at conferences and disseminated in the form of academic papers.

### Limitations

The present protocol has some limitations. First, the intensity and duration of physical activity may decrease as gestational age increases. To address this concern, physical activity monitoring will be conducted at the corresponding time point to verify compliance with activity intensity. A cut-off of 80% of baseline physical activity during the second and third trimesters will be used to determine whether compliance well. The per-protocol set will be performed as a sensitivity analysis for all patients with compliance well. Second, the rate of withdrawal may be high owing to miscarriages, preterm births, and pregnancy complications. Therefore, this study will be conducted for two years.

## 5. Conclusion

This cohort study will provide a basis for randomized controlled trials and validate a feasible, economical, safe, and non-fetotoxic approach for adjuvant blood pressure management in pregnant women with chronic hypertension.

## Supporting information

**S1 File. Study protocol original language.**
(DOCX)

**S2 File. Study protocol translation version.**
(DOCX)

## Author Contributions

**Conceptualization:** Yanxiang Lv, Tongqiang He.

**Data curation:** Yanxiang Lv, Rui Hu.

**Formal analysis:** Rui Hu, Yan Liang.

**Funding acquisition:** Yanxiang Lv.

**Investigation:** Rui Hu, Yan Liang, Ying Zhou, Yanan Lian.

**Methodology:** Yanxiang Lv, Tongqiang He.

**Project administration:** Tongqiang He.

**Writing – original draft:** Yanxiang Lv, Rui Hu.

**Writing – review & editing:** Tongqiang He.

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
