## [Decision Letter · Decision Letter 0]

25 Jun 2023

PONE-D-23-07838Daily physical activity on ambulatory blood pressure in pregnant women with chronic hypertension: A prospective cohort study protocolPLOS ONE

Dear Dr. He,

Thank you for submitting your manuscript to PLOS ONE. After careful consideration, we feel that it has merit but does not fully meet PLOS ONE’s publication criteria as it currently stands. Therefore, we invite you to submit a revised version of the manuscript that addresses the points raised during the review process.

We look forward to receiving your revised manuscript.

Kind regards,

Antonio Simone Laganà, M.D., Ph.D.

Academic Editor

PLOS ONE

Journal Requirements:

Additional Editor Comments:

The reviewers have expressed positive comments regarding your article, raising only few concerns. Considering this point, I invite authors to perform the required minor revisions.

Reviewers' comments:

Reviewer's Responses to Questions

**Comments to the Author**

1. Does the manuscript provide a valid rationale for the proposed study, with clearly identified and justified research questions?

Reviewer #1: Yes

Reviewer #2: Yes

Reviewer #3: Yes

Reviewer #4: Yes

2. Is the protocol technically sound and planned in a manner that will lead to a meaningful outcome and allow testing the stated hypotheses?

Reviewer #1: Yes

Reviewer #2: Yes

Reviewer #3: Yes

Reviewer #4: Yes

3. Is the methodology feasible and described in sufficient detail to allow the work to be replicable?

Reviewer #1: Yes

Reviewer #2: Yes

Reviewer #3: Yes

Reviewer #4: Yes

4. Have the authors described where all data underlying the findings will be made available when the study is complete?

Reviewer #1: Yes

Reviewer #2: Yes

Reviewer #3: No

Reviewer #4: No

5. Is the manuscript presented in an intelligible fashion and written in standard English?

Reviewer #1: Yes

Reviewer #2: Yes

Reviewer #3: Yes

Reviewer #4: Yes

6. Review Comments to the Author

You may also provide optional suggestions and comments to authors that they might find helpful in planning their study.

Reviewer #1: dear authors your project is absolutely valid and truly interesting

I would suggest minor revisions

Introductions

-please pay attention to the spaces

- chronic hypertensive disorders may become a serious healthcare problem with the upcoming pregnancy for the woman, in a significant percentage of cases these last may evolve to preeclampsia and even more worrying to eclampsia a condition not easy to treat I would like to at least mention this concept within the introduction (read and cite PMID: 35317697)

-regarding methods I would suggest to exclude as well patient with a subsequent diagnosis of gestational diabetes due to its additional impact on the incidence of hypertensive disorders of pregnancy

Reviewer #2: I read with great interest the Manuscript titled " Daily physical activity on ambulatory blood pressure in pregnant women with chronic hypertension: A prospective cohort study protocol" which falls within the aim of the Journal.

In my opinion, this topic analyzed is interesting enough to attract readers’ attention.

Although the manuscript can be considered already of good quality, I would suggest to take into account the following recommendations:

- Recent and novel evidence suggested that epigenetic changes, in particular altered expression of selective miRNA, may play a key role in both placental-induced diseases such as pre-eclampsia and intrauterine growth restriction. It would be mandatory to discuss (at least briefly) this topic, referring to: PMID: 28466013; PMID: 28282763.

- I suggest a round of language revision, in order to correct few typos and improve readability.

Because of these reasons, the article should be revised and completed. Considering all these points, I think it could be of interest for the readers and, in my opinion, it deserves the priority to be published after minor revisions.

Reviewer #3: This manuscript describes a protocol that will investigate the effect of daily physical activity on ambulatory blood pressure in pregnant women with chronic hypertension using a prospective cohort. One of the goals of this cohort study tends to provide the basis for randomized controlled trials, and verify an easily achieved, economical, safe, and non-fetotoxicity approach to adjuvant blood pressure management in pregnant women with chronic hypertension. I have below comments.

Since the primary outcome is the change (baseline has been used) in 24-hour ambulatory SBP from the first to the third trimester, to compare it between two groups, t-test and Mann-Whitney U test may be used instead of analysis of covariance. Similar issue exists in the analysis for secondary outcomes (blood pressure change at 2nd and 3rd trimesters).

It is not clear about the intention-to-treat and per-protocol set analysis.

What’s the expected rate of compliance of physical activity?

Reviewer #4: I read with great interest the Manuscript titled “ Daily physical activity on ambulatory blood pressure in pregnant women with chronic hypertension: A prospective cohort study protocol” (PONE-D-23-07838), which falls within the aim of this Journal. In my honest opinion, the topic is interesting enough to attract the readers’ attention. Nevertheless, authors should clarify some point and improve the discussion citing relevant and novel key articles about the topic.

Authors should consider the following recommendations:

- Manuscript should be further revised by a native English speaker

- Conclusions are not supported by data. Although the recruitment of the patients ends in December and it's specified that all relevant data from this study will be made available upon study completion, I would have appreciated if the early data had been providen

- It would be useful to know how many patients enrolled run into miscarriages, preterm birth, stillbirth and how many present a SGA

- I suggest to give informations regarding how many patients recruited for the study are affected by eclampsia and HELLP syndrome and how physical activity in women with chronic hypertensionr reduces the risk of these complications.

- Does this manuscript conform the Enhancing the QUAlity and Transparency Of health Research (EQUATOR) network guidelines? In this case they consist of SPIRIT and PRISMA-P guidelines. It would be mandatory to declare about this element.

- The real challenge in the era of molecular medicine is to find a biomarker, or evene better a panel of biomarkers, for early diagnosis of pre-eclampsia, imtrauterine growth restriction and stillbirth. I would stress this point, referring to: PMID: 28243732, PMID: 35245721.

7. PLOS authors have the option to publish the peer review history of their article (what does this mean?). If published, this will include your full peer review and any attached files.

Reviewer #1: No

Reviewer #2: **Yes: **Ilaria Cuccu

Reviewer #3: No

Reviewer #4: No

---

## [Author Response · Author response to Decision Letter 0]

8 Aug 2023

Dear Editors and Reviewers:

Thank you for your letter and the comments concerning our manuscript entitled “Daily physical activity on ambulatory blood pressure in pregnant women with chronic hypertension: A prospective cohort study protocol” (ID: PONE-D-23-07838). Those comments are all valuable for revising and improving our paper and the important guiding significance to our research. We have studied the comments carefully and have made corrections which we hope meet with approval.

Responds to Editor

Response:

Thank you for your feedback. We have carefully reviewed our manuscript and made sure that it adheres to the specified style guidelines. This includes ensuring that the file names are in accordance with PLOS ONE's requirements. We make the necessary adjustments to meet the journal's standards.

2. Please address the following queries:

Response:

We appreciate your guidance in ensuring that this information is accurately provided. This project is supported by the Key Research and Development Program of Shaanxi China (Program No. 2022SF-277) and the Scientific Research Project of Northwest Women and Children’s Hospital (Program No. 2018LQ03, 2022-YN10). 

The funders have no role in study design, data collection, analysis, decision to publish, or preparation of the manuscript.

No author receives a salary from any of the funders. 

We revised our manuscript to include this information in the appropriate section, providing a clear and comprehensive acknowledgment of the funding sources. (Lines 19-23 on pages 1-2)

Response:

We have carefully reviewed our reference list to ensure its completeness and accuracy. We have double-checked all the references cited in the manuscript and ensured that they are relevant. We did not include any retracted papers in our reference list. The changes made to the reference list during the revision process were mentioned in the response letter accompanying the revised manuscript.

We appreciate for Editors’ warm work earnestly and hope that the correction will meet with approval. 

Best regards,

Tongqiang He, MS

July 18, 2023

Department of Obstetrics and Gynecology Intensive Care Unit, Northwest Women's and Children's Hospital

Email: xbfymicu@163.com

Responds to the reviewer’s comments:

Reviewer #1: 

Dear Reviewer,

Thank you for your positive feedback and kind words regarding our project. We are delighted to hear that you find our research to be valid and interesting. Your encouragement and recognition of the value of our work mean a lot to us. We are committed to further improving and developing our project based on your constructive suggestions as follows: 

1. Please pay attention to the spaces

Response:

We apologize for any errors related to the spaces. We have diligently reviewed the entire document and made revisions accordingly throughout. We did not list the changes but marked them in blue in the revised paper.

2. Chronic hypertensive disorders may become a serious healthcare problem with the upcoming pregnancy for the woman, in a significant percentage of cases these last may evolve to preeclampsia and even more worrying to eclampsia a condition not easy to treat. I would like to at least mention this concept within the introduction (read and cite PMID: 35317697)

Response:

Thank you for your comment regarding mentioning that chronic hypertensive disorders may evolve into preeclampsia and eclampsia within the introduction. We appreciate your suggestion to consider including this important piece of information. We have revised the introduction section and included a brief mention of the potential health concerns associated with chronic hypertensive disorders during pregnancy. We have also cited the relevant paper you recommended (PMID: 35317697) as a source of further information. (Lines 67 on page 4, reference 2).

3. Regarding methods I would suggest to exclude as well patient with a subsequent diagnosis of gestational diabetes due to its additional impact on the incidence of hypertensive disorders of pregnancy

Response:

Thank you for your suggestion regarding the exclusion of patients with a subsequent diagnosis of gestational diabetes from the study. We acknowledge that these individuals may need to increase their physical activity under medical guidance to manage blood glucose levels, which could potentially influence the outcomes. We agree that excluding these individuals would help to focus the study specifically on whether moderate-intensity daily physical activity reduces ambulatory blood pressure in pregnant women with chronic hypertension. This exclusion will enhance the clarity and relevance of the results obtained. Based on your recommendation, we have revised the methods section. (Lines 154 on page 8)

We appreciate your warm work earnestly and hope that the correction will meet with approval.

Best regards,

Tongqiang He, MS

July 18, 2023

Department of Obstetrics and Gynecology Intensive Care Unit, Northwest Women's and Children's Hospital

Email: xbfymicu@163.com

Reviewer #2: 

Dear Reviewer,

Thank you for your feedback on our manuscript. We are glad to know that you find the topic interesting and believe it has the potential to attract readers' attention. We appreciate your acknowledgment of the manuscript's overall good quality.

We are also grateful for your suggestions to further enhance our work. We have carefully considered the following recommendations:

1. Recent and novel evidence suggested that epigenetic changes, in particular altered expression of selective miRNA, may play a key role in both placental-induced diseases such as pre-eclampsia and intrauterine growth restriction. It would be mandatory to discuss (at least briefly) this topic, referring to: PMID: 28466013; PMID: 28282763.

Response:

Thank you for your suggestion. We agree that discussing the role of epigenetic changes would be a valuable addition to the manuscript. We have included a brief discussion on this topic, referring to the relevant articles you have suggested (PMID: 28466013; PMID: 28282763). (Lines 277-281 on page 14, references 35 and 37)

2. I suggest a round of language revision, in order to correct few typos and improve readability.

Response:

We appreciate your suggestion of language revision, and we tried our best to improve the manuscript and enlist the help of a professional copy-editing service. Here we did not list the changes but marked them in blue in the revised paper. We have uploaded a Certificate of editing.

Thank you once again for your comments and suggestions. We token them very seriously and did our best to address them in the revised version of our manuscript.

Best regards,

Tongqiang He, MS

July 18, 2023

Department of Obstetrics and Gynecology Intensive Care Unit, Northwest Women's and Children's Hospital

Email: xbfymicu@163.com

Reviewer #3: 

Dear Reviewer,

Thank you for your comments on our manuscript. We appreciate your insights and suggestions. We have revised the manuscript according to your comment, and believe that these changes will enhance the clarity and impact of our manuscript. 

1. Since the primary outcome is the change (baseline has been used) in 24-hour ambulatory SBP from the first to the third trimester, to compare it between two groups, t-test and Mann-Whitney U test may be used instead of analysis of covariance. Similar issue exists in the analysis for secondary outcomes (blood pressure change at 2nd and 3rd trimesters).

Response:

Thank you for your feedback. We agree that the t-test and Mann-Whitney U test are more appropriate than the analysis of covariance to compare the change in 24-hour ambulatory SBP from the first to the third trimester between the two groups. We also acknowledge that these tests can be appropriate for analyzing the secondary outcomes related to blood pressure change at the 2nd and 3rd trimesters. We have made revisions to the statistical section of the manuscript. Thank you for your suggestion. (Lines 55-57on pages 3-4, Lines 259-261 on page 13)

2. It is not clear about the intention-to-treat and per-protocol set analysis.

Response:

We apologize for not providing clarity regarding the intention-to-treat and per-protocol set analysis in our manuscript. In the revised version, we have included a detailed explanation of these analysis methods. The intention-to-treat population will comprise all patients, except for those who do not complete ambulatory blood pressure measurement and physical activity monitoring at baseline or in the third trimester for any reason, such as miscarriage or preterm birth. The per-protocol set was used for sensitivity analysis. The per-protocol set will comprise patients who complete ambulatory blood pressure measurements and physical activity monitoring at three time points with good compliance. We appreciate your feedback and have addressed this issue accordingly. (Lines 262-267 on page 13)

3. What’s the expected rate of compliance of physical activity?

Response:

The expected rate of compliance for physical activity varies depending on several factors, such as physiologic responses to physical activity during pregnancy, gestational age, and complications of pregnancy. It is difficult to provide a specific rate due to the limited research available on physical activity in pregnant women with chronic hypertension. A prospective cohort study [1] was conducted to estimate physical activity in healthy women before and during pregnancy, and results showed that the percentage of women who complied with the physical activity recommendations was 64.4% in the pre-pregnancy stage, 54.5%, 61.9%, 59.1% in the first, second, and third trimesters. A 2-center, single-blinded randomized clinical trial [2] was conducted to determine whether an aerobic exercise training intervention reduces ambulatory BP among patients with resistant hypertension, showing that the median adherence rate in the exercise arm was 98.8%. We plan to address this question in pregnant women with chronic hypertension by closely monitoring participant adherence to physical activity and conducting follow-up assessments to determine compliance rates.

References

[1] Román-Gálvez MR, Amezcua-Prieto C, Salcedo-Bellido, I, et al. Physical activity before and during pregnancy: A cohort study. INT J GYNECOL OBSTET. 2021-03-01;152(3):374-381.

[2] Lopes S, Mesquita-Bastos J, Garcia C, et al. Effect of exercise training on ambulatory blood pressure among patients with resistant hypertension: a randomized clinical trial. JAMA Cardiol. 2021;6(11):1317-23.

Once again, we thank you for your thoughtful comments and for your time in reviewing our work.

Best regards,

Tongqiang He, MS

July 18, 2023

Department of Obstetrics and Gynecology Intensive Care Unit, Northwest Women's and Children's Hospital

Email: xbfymicu@163.com

Reviewer #4:

Dear Reviewer,

Thank you for your feedback on our manuscript. We appreciate your positive comments regarding the relevance and interest of our topic. We also acknowledge your suggestion to clarify certain points and improve the discussion by citing relevant and novel key articles. To address this concern, we revised the manuscript following your recommendations: 

1. Manuscript should be further revised by a native English speaker

Response

We appreciate your suggestion and understand the importance of ensuring the language quality of the manuscript. We have revised the manuscript again and have sought the assistance of a professional native English speaker for further language editing. And here we did not list the changes but marked them in blue in the revised paper. 

2. Conclusions are not supported by data. Although the recruitment of the patients ends in December and it's specified that all relevant data from this study will be made available upon study completion, I would have appreciated if the early data had been providen.

Response

Thank you very much for your interest in our study and your valuable suggestion. At this stage, we are still in the process of recruitment and data collection for our study. Due to the long duration of each participant's study period, from the first trimester to the termination of pregnancy which takes approximately around 7 months, the number of participants who have completed the study is relatively small. We currently do not have enough data to support reliable conclusions. We place great importance on the rigor of scientific research, and as such, we will complete the study according to our planned schedule and provide all relevant data once the study is finished. As the study progresses, we will continue to collect more data and ensure consistency between the final conclusions and the data collected.

We are so sorry for being temporarily unable to provide early data considering the reliability of the conclusions. Thank you for your understanding and support. 

3. It would be useful to know how many patients enrolled run into miscarriages, preterm birth, stillbirth and how many present a SGA

Response

Thank you for your suggestion. We agree that it would be useful to provide information about the number of patients who experience miscarriages, preterm birth, stillbirth, and those who present with small for gestational age (SGA). We revised the manuscript and added the number and incidence of adverse maternal and fetal outcomes as exploratory outcomes, including miscarriages, preterm birth, stillbirth, and small for gestational age. (Lines 228-230 on page 11)

4. I suggest to give informations regarding how many patients recruited for the study are affected by eclampsia and HELLP syndrome and how physical activity in women with chronic hypertension reduces the risk of these complications.

Response

We appreciate your suggestion to include information about how many patients recruited for the study are affected by eclampsia and HELLP syndrome, as well as how physical activity in women with chronic hypertension reduces the risk of these complications. We addressed eclampsia and HELLP syndrome as exploratory outcomes in our revised manuscript. (Lines 228-230 on page 11)

5. Does this manuscript conform the Enhancing the QUAlity and Transparency Of health Research (EQUATOR) network guidelines? In this case they consist of SPIRIT and PRISMA-P guidelines. It would be mandatory to declare about this element.

Response

We sincerely apologize for not explicitly mentioning conforming to the Enhancing the Quality and Transparency of Health Research (EQUATOR) Network guidelines in our manuscript. We acknowledge the importance of adhering to these guidelines to ensure the transparency and quality of our research. In our revised version, we included a statement declaring our adherence to these guidelines and uploaded a checklist in the submission system. This will help to address any concerns regarding the completeness and transparency of our research methodology and reporting. (Lines 134-137 on page 5)

6. The real challenge in the era of molecular medicine is to find a biomarker, or evene better a panel of biomarkers, for early diagnosis of pre-eclampsia, imtrauterine growth restriction and stillbirth. I would stress this point, referring to: PMID: 28243732, PMID: 35245721.

Response

Thank you for your comment. We completely agree with you at this point. We have revised the background section of manuscript and referred to the relevant articles you have suggested (PMID: 28243732, PMID: 35245721). (Lines 71-73 on page 4, references 7 and 9)

We appreciate your warm work earnestly and hope that the correction will meet with approval.

Best regards,

Tongqiang He, MS

July 18, 2023

Department of Obstetrics and Gynecology Intensive Care Unit, Northwest Women's and Children's Hospital

Email: xbfymicu@163.com

---

## [Decision Letter · Decision Letter 1]

5 Dec 2023

Effect of daily physical activity on ambulatory blood pressure in pregnant women with chronic hypertension : A prospective cohortstudy protocol

PONE-D-23-07838R1

Dear Dr. He,

We’re pleased to inform you that your manuscript has been judged scientifically suitable for publication and will be formally accepted for publication once it meets all outstanding technical requirements.

Kind regards,

Nur Aizati Athirah Daud, Ph.D.

Academic Editor

PLOS ONE

Additional Editor Comments (optional):

Reviewers' comments:

Reviewer's Responses to Questions

**Comments to the Author**

1. Does the manuscript provide a valid rationale for the proposed study, with clearly identified and justified research questions?

Reviewer #1: Yes

Reviewer #3: Yes

Reviewer #5: Yes

2. Is the protocol technically sound and planned in a manner that will lead to a meaningful outcome and allow testing the stated hypotheses?

Reviewer #1: Yes

Reviewer #3: Yes

Reviewer #5: Yes

3. Is the methodology feasible and described in sufficient detail to allow the work to be replicable?

Reviewer #1: Yes

Reviewer #3: Yes

Reviewer #5: Yes

4. Have the authors described where all data underlying the findings will be made available when the study is complete?

Reviewer #1: Yes

Reviewer #3: Yes

Reviewer #5: Yes

5. Is the manuscript presented in an intelligible fashion and written in standard English?

Reviewer #1: Yes

Reviewer #3: Yes

Reviewer #5: Yes

6. Review Comments to the Author

You may also provide optional suggestions and comments to authors that they might find helpful in planning their study.

Reviewer #1: Dear authors

thank you for addressing all comments

the paper has improved so i will suggest it to be accepted

Reviewer #3: My comments have been addressed. I don't have further comments.

Reviewer #5: This is a well prepared study protocol to elucidate the effects of physical activity on pregnant patients with chronic hypertension. I think with the revision performed according to the other reviewers' comments the study protocol is suitable for publication in the current form.

7. PLOS authors have the option to publish the peer review history of their article (what does this mean?). If published, this will include your full peer review and any attached files.

Reviewer #1: No

Reviewer #3: No

Reviewer #5: No

---

## [Editor Report · Acceptance letter]

2 Jan 2024

PONE-D-23-07838R1 

PLOS ONE

Dear Dr. He, 

I'm pleased to inform you that your manuscript has been deemed suitable for publication in PLOS ONE. Congratulations! Your manuscript is now being handed over to our production team.

Kind regards, 

on behalf of

Dr. Nur Aizati Athirah Daud 

Academic Editor

PLOS ONE